# Transposable element expression is associated with sex chromosome number in humans

Jordan Teoli[1,2,3,4*], Miriam Merenciano[2], Marie Fablet[2,5], Anamaria Necsulea[2], Daniel Siqueira-de-Oliveira[2], Alessandro Brandulas-Cammarata[6,7], Audrey Labalme[8], Hervé Lejeune[9], Jean-François Lemaitre[2], François Gueyffier[2], Damien Sanlaville[8], Claire Bardel[1,2], Cristina Vieira[2�], Gabriel AB Marais[2,4,10,11�], Ingrid Plotton[1,3,9�]

**1** Laboratoire de Biochimie et Biologie Moléculaire, Centre de Biologie et Pathologie Est, Hospices Civils de Lyon, Bron, France, **2** Laboratoire de Biométrie et Biologie Evolutive, UMR 5558, CNRS, Universite Claude Bernard Lyon 1, Villeurbanne, France, **3** Institut Cellule Souche et Cerveau (SBRI), Unité INSERM 1208, Centre de Recherche INSERM, Universite Claude Bernard Lyon 1, Bron, France, **4** CIBIO, Centro de Investigação em Biodiversidade e Recursos Genéticos, InBIO Laboratório Associado, Campus de Vairão, Universidade do Porto, Vairão, Portugal, **5** Institut Universitaire de France, Paris, France, **6** Department of Ecology and Evolution, University of Lausanne, Lausanne, Switzerland, **7** SIB Swiss Institute of Bioinformatics, Lausanne, Switzerland, **8** Service de Génétique, Hospices Civils de Lyon, Bron, France, **9** Service de Médecine et Biologie de la Reproduction, Hôpital Femme-Mère-Enfant, Hospices Civils de Lyon, Bron, France, **10** Departamento de Biologia, Faculdade de Ciências, Universidade do Porto, Porto, Portugal, **11** BIOPOLIS Program in Genomics, Biodiversity and Land Planning, CIBIO, Campus de Vairão, Vairão, Portugal

☀ Equal contributions as senior authors
* jordan.teoli@chu-lyon.fr

## Abstract

Why women live longer than men is still an open question in human biology. Sex chromosomes have been proposed to play a role in the observed sex gap in longevity, and the Y male chromosome has been suspected of having a potential toxic genomic impact on male longevity. It has been hypothesized that transposable element (TE) repression declines with age, potentially leading to detrimental effects such as somatic mutations and disrupted gene expression, which may accelerate the aging process. Given that the Y chromosome is rich in repeats, age-related increases in TE expression could be more pronounced in males, likely contributing to their reduced longevity compared to females. In this work, we first studied whether TE expression is associated with the number of sex chromosomes in humans. We analyzed blood transcriptomic data obtained from individuals of different karyotype compositions: 46,XX females (normal female karyotype), 46,XY males (normal male karyotype), as well as males with abnormal karyotypes, such as 47,XXY, and 47,XYY. We found that sex chromosomes might be associated to TE expression, with the presence and number of Y chromosomes particularly associated with a global increase in TE expression. This tendency was also observed across several TE subfamilies. We also tested whether TE expression is higher in older males than in older females using published human blood transcriptomic data from the Genotype-Tissue Expression (GTEx) project. However, we did not find increased TE expression

**Data availability statement:** Fastq files of the gonosome aneuploidy dataset generated in this study has been deposited at the European Genome-phenome Archive (EGA), which is hosted by the EBI and the CRG, under accession number EGAD00001011202 (https://ega-archive.org/datasets/EGAD00001011202). Scripts and code were described in detail in the Materials and methods section. The code of the pipeline used to generate gene and TE count files from FastQ files is available here: https://github.com/teolijo/longevitY/blob/main/pipeline. Numerical data underlying figures and tables are available here: https://dx.doi.org/10.6084/m9.figshare.28365464.

**Funding:** This work was funded by an internal collaborative grant of Laboratoire de biométrie et biologie évolutive and by ANR (grant number ANR-20-CE02-0015 to CV). The funders had no role in study design, data collection and analysis, decision to publish, or preparation of the manuscript.

**Competing interests:** The authors have declared that no competing interests exist.

in older males compared to older females probably due to the heterogeneity of the dataset. Our findings suggest an association between sex chromosome content and TE expression and open a new window to study the toxic effect of the Y chromosome in human longevity.

## Author summary

Why women live longer than men is still an open question in human biology. Sex chromosomes have been proposed to play a role in the observed sex gap in longevity, and the Y male chromosome has been suspected of having a potential toxic genomic impact on male longevity. It has been hypothesized that in aging individuals, transposable element (TE) repression is diminished, which could lead to detrimental effects (e.g., somatic mutations, perturbed gene expression) and to an acceleration of the aging process. The Y chromosome is typically enriched in repetitive elements that can increase their expression with age in males, reducing their longevity compared to females. In this work, we use for the first time transcriptomic data from humans with atypical karyotypes to associate sex chromosome content and TE expression. Particularly, we observed that the presence and number of Y chromosomes was associated with increased TE expression. These findings open a new window to study the existence of a toxic Y effect on men's longevity.

## Introduction

Differences in longevity between sexes are prevalent across the tree of life, a phenomenon known as sex gap in longevity (SGL) [1]. In both vertebrates and invertebrates, the heterogametic sex (males in XY, and females in ZW systems) usually has a reduced longevity compared to the homogametic sex [1]. SGL is also observed in humans, where on average, life expectancy at birth is five years longer for women than for men [2]. The female-biased gap in longevity is consistently observed in nearly all human populations [3], and it explains the increased prevalence of women among supercentenarians [4].

Although cultural factors favor the extended longevity in women, several biological hypotheses have also been proposed to underlie the SGL phenomenon, which are not mutually exclusive [3,5]. First, the sex-specific production of hormones (e.g., androgens) has been suggested to contribute to sexual dimorphism in longevity. In humans, it was found that the removal of sex-specific hormones increases male longevity [6]. Second, due to the exclusive maternal transmission of mitochondria, natural selection cannot target deleterious mutations in the mitochondrial genome that specifically impact male fitness. These mutations can thus freely accumulate and may reduce male longevity, a phenomenon called the mother's curse [6–8]. Third, since the heterogametic sex carries one copy of each sex chromosome (e.g., XY

in mammals and ZW in birds), it is consequently more susceptible to recessive X-linked (or Z-linked) deleterious genetic mutations compared to the homogametic sex (e.g., XX in mammals and ZZ in birds), as proposed by the unguarded X hypothesis [9]. Both theory and empirical data suggest that the latter mechanism explains a part of the SGL [10–12].

Finally, the toxic Y hypothesis has been recently proposed to explain the existence of SGL. This hypothesis relies on the high repetitive element content in Y (or in W, the equivalent of the Y chromosome in the ZW sex-determination system) chromosomes (reviewed in [11,13]. Y (or W) chromosomes frequently exhibit lower rates of recombination, leading to an accumulation of repetitive sequences, including transposable elements (TEs), on these chromosomes [14]. TEs are DNA sequences with the ability to move within the genome, highly enriched in heterochromatin, and that contribute to approximately 45% of the human genome sequence [15–17].

While many TEs have a neutral impact, certain insertions can interfere with gene function or result in detrimental chromosomal rearrangements [18,19]. Epigenetic silencing mechanisms exist to prevent TE expression and transposition, like DNA methylation, histone modifications, and the production of small interfering RNAs [20]. However, aging has been shown to disrupt this epigenetic regulation at constitutive heterochromatin causing an increased TE activation in several organisms including humans [21–28], thus enhancing the probability to induce detrimental effects through somatic mutations or other mechanisms (e.g., accumulation of TE transcripts into the cytoplasm, interaction between TE transcripts and genes/proteins) [29–31]. The toxic Y hypothesis thus suggests that more TEs might become active in males compared to females, generating more detrimental effects, accelerating aging and likely reducing longevity in males.

This hypothesis has already been investigated in the fruit fly *Drosophila melanogaster*, where TE expression was found to be higher in older males compared to older females [28]. This observation was associated with a loss of heterochromatin in repetitive elements during aging in male flies [28]. Moreover, flies with additional Y chromosomes showed decreased longevity, further suggesting that the number of Y chromosomes influences organismal survival in this species females [28]. However, another recent study using flies with variable levels of heterochromatin in the Y chromosome pointed out that the presence, number, or size of the Y chromosome has no impact on sexual dimorphism in longevity [32]. Hence, the harmful effects of the Y chromosome on longevity are still under debate.

In humans, men with 47,XYY and 47,XXY abnormal karyotypes (with an extra Y or X chromosome, respectively) have reduced longevities: 47,XYY men were associated with a 10-year reduction in longevity while 47,XXY men were associated with a two-year reduction compared to 46,XY individuals [33,34]. This was observed despite the extra Y having smaller observable effects on 47,XYY individual's biology and health than the extra X in 47,XXY individuals, which results in the Klinefelter syndrome [33,34]. In humans, the Y chromosome is relatively small (~57 Mb) compared to the X chromosome (~156 Mb) [35] and is also richer in repetitive elements (85% of the base pairs of the Y chromosome) compared to the rest of the genome (53.9%) [15–17]. However, the human Y chromosome is not enriched in TE sequences (31.8%) [17] compared to the rest of the genome (46.0%) [15,16]. Yet, in humans, information is lacking about the effect of sex chromosomes on TE expression, the contribution of TEs in longevity, and thus the possible toxic Y effect in this species.

In this work, we first aimed to assess the effect of sex chromosomes on TE expression, and more specifically to test whether the presence and number of Y chromosomes were associated with a higher TE expression. We analyzed transcriptomic data from individuals with different karyotypes, including 46,XX females (normal female karyotype), 46,XY males (normal male karyotype), as well as males with atypical karyotypes, such as 47,XXY and 47,XYY. Expression of TE was compared between karyotypes and the comparisons between 47,XYY and 46,XY and between 47,XXY and 46,XY are particularly relevant to assess the effect of one supplementary Y chromosome and one supplementary X chromosome on TE expression from the same phenotypic sex background. We found an association between sex chromosome content and TE expression, particularly that the presence and number of Y chromosomes might be linked to increased overall TE expression. This tendency was also observed for several TE subfamilies. We also tested whether there is an increased TE expression in older males compared to older females by using published transcriptomic data. However, we did not find an increased TE expression in older males compared to older females probably due to the heterogeneity of the dataset.

Overall, this work suggests an association between sex chromosome content and TE expression and opens a new window to study the toxic effect of the Y chromosome in human longevity.

## Results

### Males with sex aneuploidies (47,XXY and 47,XYY) have higher transcriptomic differences than 46,XY males compared to 46,XX females

To investigate what is the impact of sex chromosomes on TE expression, we generated RNA-seq data from blood samples of males (46,XY) and females (46,XX) from the Lyon University Hospital. In addition, we also generated RNA-seq data from males with different sex chromosome aneuploidies: males with Klinefelter syndrome harboring an additional X chromosome (47,XXY), and males with Jacob syndrome harboring an additional Y chromosome (47,XYY). In total, we obtained blood samples from 24 individuals (six females 46,XX, six males 46,XY, eight males 47,XXY, and four males 47,XYY) (Table A in S1 File). Pairwise Wilcoxon tests revealed a significant difference in age distribution only between 46,XX and 47,XXY individuals (p-value = 0.033). This finding suggests that age could act as a confounding factor in the relationship between TE expression and karyotype. However, it is worth noting that any such conclusion would introduce a conservative bias when evaluating the correlation between TE expression and the number of Y chromosomes in a karyotype. Notably, individuals with 47,XXY (mean age 23.4 years) and 47,XYY (mean age 29.5 years) tend to be younger compared to those with 46,XX (mean age 39.2 years) and 46,XY (mean age 41.5 years) karyotypes.

We first analyzed gene expression to see whether different karyotypes have changes in their transcriptomic profile (Fig A in S1 File) (see **Materials and Methods**). Principal component analysis (PCA) separates female samples (46,XX) from male samples (with and without aneuploidies) (PC1: 29% of the variance) (Fig B in S1 File). Besides that, we found that the number of significantly differentially expressed genes (DEGs) varied depending on the pair of karyotypes compared (Fig C and Table B in S1 File). Comparing 46,XY males with 46,XX females, 19 (0.10%) and 18 (0.09%) genes were found to be upregulated and downregulated, respectively, among the 19,947 studied genes (Table B in S1 File). We found that while most of the upregulated genes (13/19, 68.42%) were Y-linked genes, most of the downregulated ones (14/18, 77.78%) were X-linked genes (Fig D and Tables B-D in S1 File and S1 and S2 Data). Furthermore, we found an increased number of DEGs when comparing samples from individuals with sex chromosome aneuploidies (47,XXY and 47,XYY males) to female (46,XX) than to male (46,XY) samples. Indeed, 47,XXY and 47,XYY males showed 1,893 (9.49%) and 1,510 (7.57%) DEGs compared to 46,XX females, respectively. However, 47,XXY and 47,XYY males only showed 141 (0.71%) and 58 (0.29%) DEGs compared to 46,XY males, respectively (Fig C and Table B in S1 File and S2 Data). We also found a small amount of DEGs when comparing 47,XYY and 47,XXY males (50, 0.25%) (Fig C and Table B in S1 File).

Finally, Gene Ontology (GO) and pathway enrichment analysis found the largest number of significant terms when 47,XXY and 47,XYY were compared to 46,XX but few considering the four other comparisons (S3 Data). Overall, terms associated with demethylase activity, cell cycle, and in protein or DNA production were the most shared between the six pairwise karyotype comparisons and between the databases used (GO, KEGG or Reactome) (Figs E-G in S1 File). Interestingly, biological pathways concerning senescence, like DNA damage/telomere stress induced senescence, senescence-associated secretory phenotype, and oxidative stress induced senescence, were found downregulated in 47,XXY karyotype compared to 46,XX, 46,XY and 47,XYY karyotypes (Fig H in S1 File and S3 Data). Furthermore, some of the DEGs were also previously found in other studies (see Text A in S1 File).

Altogether, we observed that males with 47,XYY and 47,XXY aneuploidies exhibit greater disparities in gene transcriptomic profiles compared to 46,XX females, in contrast to the comparison between 46,XY males and 46,XX females.

## The number of Y chromosomes tend to be associated with an increased overall TE expression

To check whether the sex chromosome content is associated with overall increased levels of TE expression, we measured TE transcript amounts using the *TEcount* module of *TEtools* [36] in all the RNA samples (Fig A in S1 File).

PCA analysis, based on normalized counts of TE subfamilies, showed two different clusters: one for female samples (46,XX) and another for male samples (46,XY, 47,XXY, or 47,XYY); similar to what we observed in gene expression analysis (Fig I in S1 File).

We assessed the effect of an extra Y or X chromosome on overall TE transcript levels by first comparing 47,XYY and 47,XXY individuals to 46,XY males, respectively. We expected TE expression to increase with the presence of an additional sex chromosome, particularly with an extra Y chromosome. Pairwise comparisons showed no statistically significant increase in TE expression in any of the comparisons (Wilcoxon test: p-values > 0.05), although as expected there was a tendency to find more TE expression in individuals with atypical karyotypes, and specially in 47,XYY males (Fig 1A). Similarly, no statistically significant increase in TE transcript amounts was observed between 46,XY males and 46,XX females (Wilcoxon test: p-value = 0.310) (Fig 1A), although 46,XY males tended to have an overall increased TE transcript levels.

We then checked the effect of an extra sex chromosome on TE transcript levels compared to 46,XX females. This time, we found a statistically significant overall increase in TE transcript levels between 46,XX females and 47,XXY males (Wilcoxon test: p-value = 0.029) and 47,XYY males (Wilcoxon test: p-value = 0.010) (Fig 1A).

In summary, we observed that the presence of an additional sex chromosome tended to increase the overall amounts of TE transcripts, and particularly the presence of an extra Y chromosome (Fig 1A). These results suggested that the presence of the Y chromosome might be associated with an increase in TE transcript levels, as postulated in the toxic Y hypothesis.

We performed the same analysis as before focusing on five TE classes/orders/superfamilies (LTR order, SVA superfamily, SINE superfamily, LINE superfamily, and DNA transposon class) (Fig 1B–F).

The presence of an additional Y chromosome in 47,XYY males significantly increased the expression of LTR and SVA elements compared to 46,XY males (Wilcoxon test: p-value = 0.019 for LTR, and p-value = 0.038 for SVA) (Fig 1B and 1C). However, no significant difference was observed for SINE, LINE, and DNA transposons, although there was a trend suggesting increased TE expression in 47,XYY males relative to 46,XY males (Fig 1D–F). On the other hand, the presence of an additional X chromosome in 47,XXY males did not significantly modify TE expression of any of the five classes/orders/ superfamilies compared to 46,XY males (Wilcoxon test: p-value > 0.05) (Fig 1B–F). There was also no statistically significant difference in TE expression between 46,XY males and 46,XX females in any of the five classes/orders/superfamilies. (Fig 1B–F).

Extensive pairwise karyotype comparisons also showed that the presence of additional sex chromosomes increased the expression of some TE elements when compared to 46,XX females. Indeed, we found a statistically significant increase in TE expression considering LTR, SVA, SINE, and LINE elements in 47,XYY males compared to 46,XX females (Wilcoxon test: p-values = 0.010, 0.010, 0.010, and 0.038 respectively (Fig 1B–E) and considering LTR, SVA, and SINE elements in 47,XXY males compared to 46,XX females (Wilcoxon test: p-value = 0.003, 0.001, 0.010, respectively) (Fig 1B–D).

Then, we focused on TE groups that are known to be transcriptionally active in humans: SVA superfamily, HERVK family, AluS group, L1 family, and AluY group [37]. Some of these elements (SVA and HERVK) showed a significant increase on TE transcript levels in 47,XYY males compared to 46,XY males. None of these elements showed a significant difference on TE transcript levels in 47,XXY males compared to 46,XY males (Fig J in S1 File).

These findings suggest that an additional sex chromosome, particularly the Y chromosome, may be linked to increased TE expression in a class-specific manner. Moreover, the presence of a Y chromosome may specifically enhance the expression of some active TEs.

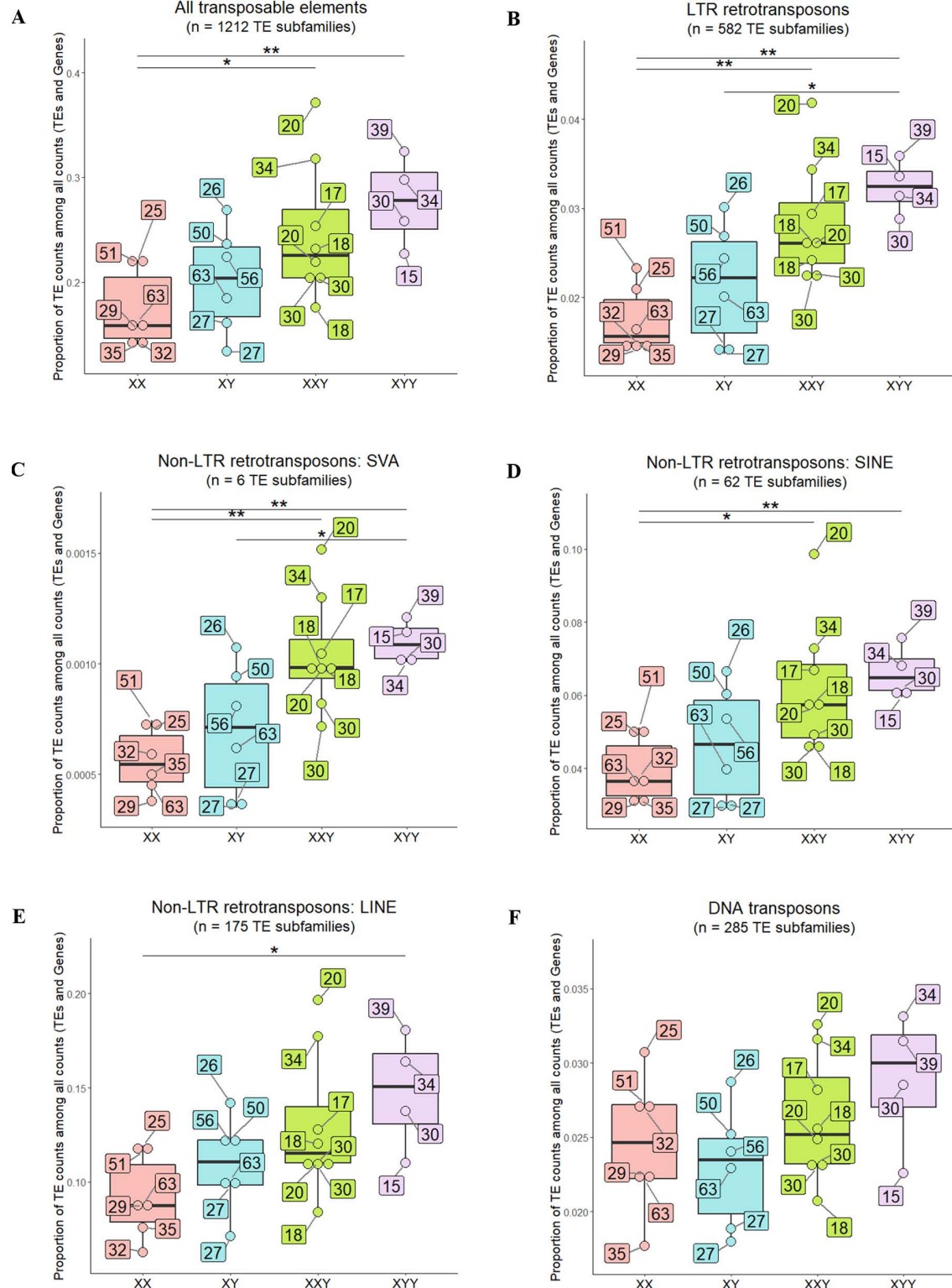

**Fig 1. TE expression in the different karyotypes after removing batch effect, considering (A) all TE subfamilies (1,212 TE subfamilies) and (B) TE subfamilies belonging to LTR retrotransposons (582 TE subfamilies), (C) SVA non-LTR retrotransposons (6 TE subfamilies), (D) SINE non-LTR retrotransposons (62 TE subfamilies), (E) LINE non-LTR retrotransposons (175 TE subfamilies), or (F) DNA elements (285 TE subfamilies).**

Global TE expression was measured in each karyotype (x-axis) as the proportion of TE read counts among all read counts (TEs and genes) (y-axis). In this calculation, read counts were used after DESeq2 normalization ("normalized counts") to remove any depth sequencing bias. Each dot represents one individual. Dots are colored according to karyotype (four levels: XX, XY, XXY, XYY). The age of each individual is indicated in a labeled box linked to the corresponding dot. Batch effect was removed before graphical display. P-values from Wilcoxon test comparing pairwise karyotypes: (*) p-value < 0.05 and (**) p-value < 0.01..

Then, considering TE subfamilies, none of them were differentially expressed between 47,XYY or 47,XXY and 46,XY males. Some TE subfamilies were found differentially expressed when 46,XY, 47,XXY or 47,XYY males were compared to 46,XX females (Fig K and Table E in S1 File), the majority belonging to ERV1, ERVK and ERVL families from the LTR order (S4 and S5 Data). Despite this, for the vast majority of differentially expressed TE subfamilies, we found a tendency to find 47,XYY individuals being the ones with the highest TE expression, 46,XX individuals being the ones with the lowest, and 47,XXY and 46,XY individuals the ones with an inbetween TE expression (Figs 2 and L in S1 File).

Variation of expression between two karyotypes of each TE subfamily is represented by a log2 fold-change (log2FC) (i.e., log base 2 of the ratio of predicted normalized counts between one karyotype to another). Thus, comparing the

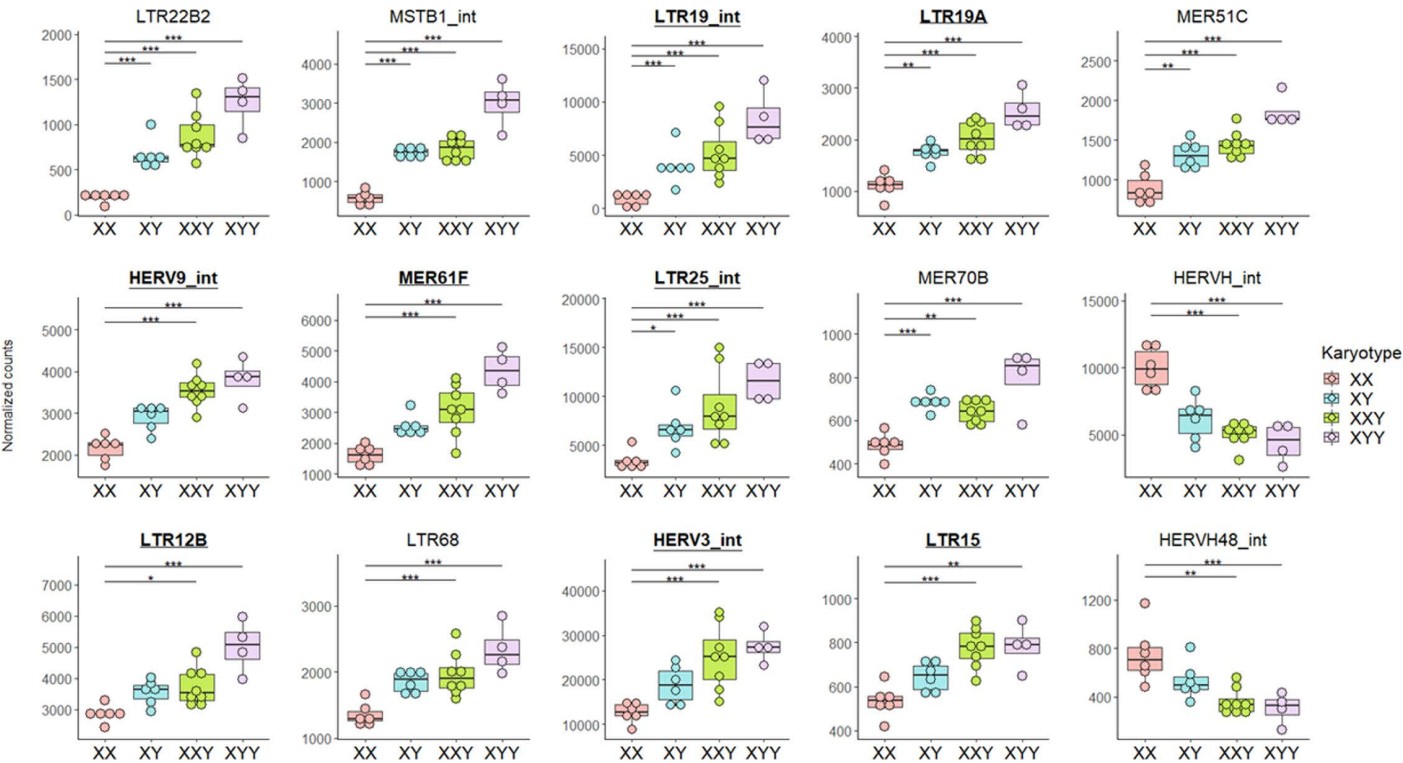

**Fig 2. Boxplots for the 15 most significantly differentially expressed TE subfamilies according to karyotype using the likelihood-ratio test, after removing batch effect.** Specific TE subfamily expression was estimated using the number of read counts after DESeq2 normalization ("normalized counts") to remove any depth sequencing bias. Each dot represents one individual. Dots are colored according to karyotype (four levels: XX, XY, XXY, XYY). Batch effect was removed before graphical display. Adjusted p-values from DESeq2: (*) p-value < 0.05, (**) p-value < 0.01, (***) p-value < 0.001. All of these 15 subfamilies had at least one copy on the Y chromosome, except for HERVH48-int, in the TE sequence reference file. All of these 15 subfamilies had at least one copy on the X chromosome in the TE sequence reference file. Underlined bold TE subfamilies are enriched in the Y chromosome (binomial test adjusted p-value < 0.05 and number of observed copies located on Y chromosome> expected). See Fig L in S1 File for the other significantly differentially expressed TE subfamilies according to karyotype. See S5 Data for number of TE subfamilies upregulated and downregulated per TE family or TE superfamily/order/class in all the pairwise karyotype comparisons.

median log2 fold-change of all TE subfamilies to zero, we can assess if an overall difference in TE expression between two karyotypes exists. We found that the median log2 fold-change of TE subfamilies was not significantly different from zero in 47,XYY males compared to 46,XY males (median = -0.002, Wilcoxon test: p-value = 0.926) (Fig K in S1 File). In contrast, the median log2 fold-change of TE subfamilies, although low, was significantly greater than zero in 47,XXY males compared to 46,XY males (median = 0.015, p-value = 0.008). Finally, we observed that the median log2 fold-change of TE subfamilies was significantly greater than zero in 46,XY males compared to 46,XX females, indicating an overall upregulation of TE subfamilies in the former (median = 0.091, p-value < 0.001) (Fig K in S1 File).

Then, we estimated which TE subfamilies were enriched in the Y chromosome among the 1,246 TE subfamilies studied herein (S6 Data). We found that 9.07% (113/1,246) of all TE subfamilies were enriched in the Y chromosome (binomial test adjusted p-value < 0.05 and number of observed copies located on Y chromosome > expected). Eight TE subfamilies out of the 15 most significant ones whose expression depended on the karyotype (53.33%) were enriched in the Y chromosome and that was greater than expected (p-value < 0.001) (Fig 2). Among the upregulated TE subfamilies found when we compared 46,XY, 47,XXY, or 47,XYY to 46,XX individuals, the proportion of TE subfamilies enriched in the Y chromosome (42.86% (3/7), 26.58% (21/79), and 37.21% (16/43) respectively) were also greater than expected (p-values = 0.020, < 0.001, and < 0.001, respectively) (Table E in S1 File).

Furthermore, we found a significant increased proportion of TE transcripts in 47,XYY males compared to 46,XY males considering TE subfamilies with no copies carried by the Y chromosome, with no copies carried by the X chromosome, or with no copies carried by either the Y or X chromosomes (Wilcoxon test: p-values = 0.019, 0.010, and 0.038 respectively) and no significant difference in proportion of TE transcripts considering Y-enriched, Y-depleted, neither Y-enriched nor Y-depleted TE subfamilies (Wilcoxon test: p-values = 0.114, 0.067, and 0.067 respectively). When the 47,XXY males were compared to 46,XY males no significant difference in the proportion of TE transcripts were noted considering separately expression of Y-enriched, Y-depleted, neither Y-enriched nor Y-depleted TE subfamilies, or TE subfamilies with no copies carried by the Y chromosome, the X chromosome or neither (Wilcoxon test: p-values = 0.345, 0.345, 0.345, 0.228, 0.108, and 0.142 respectively) (Fig M in S1 File). The Y chromosome may increase the expression of Y-linked TEs but also not Y-linked TEs.

In summary, these results might suggest that TE expression is dependent on the sex chromosomes contained in the karyotype. Particularly, the presence and number of Y chromosomes might be associated with increased TE expression. Indeed, the majority of significant increase in TE expression were found by comparing 47,XYY males to 46,XY males (one extra Y chromosome) and when 47,XYY, 47,XXY, or 46,XY males were compared to 46,XX females (presence of Y chromosome(s) versus absence) while almost no significant difference in TE expression was noted by comparing 47,XXY males to 46,XY males (one extra X chromosome).

## 47,XYY individuals have an enrichment of upregulated TE subfamilies in upstream region of upregulated genes

To see if there was any correlation between differentially expressed TE subfamilies and DEGs, we tested whether insertions from differentially expressed TE subfamilies are enriched or depleted in DEGs upstream regions (3 kb from the transcription start site). To that end, we performed permutation tests for each pairwise karyotype comparison in both upregulated and downregulated genes separately (see **Materials and Methods**). Results revealed an enrichment of TEs from upregulated subfamilies in upstream regions of upregulated genes exclusively in 47,XYY compared to 46,XX karyotype (p-value = 0.017). In contrast, we found a depletion of TE copies from upregulated TE subfamilies in the upstream region of downregulated genes when comparing 47,XYY to 46,XX karyotype (p-value = 0.009). We then used the same approach focusing on TE copies from downregulated TE subfamilies. We only found an enrichment of TE copies in the upstream region of downregulated genes in 47,XYY compared to 46,XX karyotype (p-value = 0.021).

Taken together, we found enrichment of differentially expressed TE subfamilies in the upstream regions of differentially expressed genes only in the 47,XYY vs 46,XX comparison.

## Some TE transcripts could come from passive co-transcription with genes

In addition to originating from autonomous expression, TE transcripts could also arise from passive co-transcription with genes, such as through intron retention or pervasive intragenic transcription (i.e., non-coding transcription) [38]. In fact, the majority of TE-derived RNA-seq reads typically stem from co-transcription or pervasive transcription [39,40]. Hence, we checked whether some TE transcripts come from gene transcription rather than autonomous transcription and could contribute to some TE subfamily differentially expressed.

We tested whether TE copies from upregulated subfamilies were enriched within upregulated genes for each pairwise karyotype comparison. Then, we investigated whether TE copies from downregulated subfamilies were enriched within downregulated genes for each pairwise karyotype comparison. Comparing 46,XY males to 46,XX females, we found two upregulated genes containing one or more TE insertions from upregulated TE subfamilies, but zero overlap between downregulated genes and TE copies from downregulated TE subfamilies (S7 Data). Considering the comparison 47,XXY versus 46,XX, these numbers rose to 228 and 88 respectively. There were 71 and 123 considering the comparison 47,XYY versus 46,XX (S7 Data). Except when the number of overlaps between genes and TE was equal to zero, we found a higher proportion of upregulated genes containing at least one copy of one of the upregulated TE subfamilies and a higher proportion of downregulated genes containing at least one copy of one of the downregulated TE subfamilies in 46,XY versus 46,XX, 47,XXY versus 46,XX, and 47,XYY versus 46,XX comparisons (Table F in S1 File).

In summary, these results might suggest that some TE transcripts could come from passive co-transcription with genes.

## No significant association of TE expression in blood cells with age

The toxic Y hypothesis also relies on the fact that aging disrupts the mechanisms of epigenetic regulation at constitutive heterochromatin, consequently leading to heightened activation of TE [21–28]. We investigated whether age impacts TE expression differently between males and females, hypothesizing that older males exhibit more TE expression than older females. As such, we used published RNA-seq data from the Genotype-Tissue Expression (GTEx) project. To minimize data heterogeneity and to maintain consistency with our previous analysis, we selected blood samples exclusively from non-Latino and non-Hispanic individuals without any history of persistent viral infection or dementia (Fig N in S1 File) (see **Material and Methods**). We thus compared overall TE expression in males and females belonging to five different age groups from 20 to 70 years old. We divided the dataset in two groups: one excluding data obtained from patients with current or prior histories of cancer or cardiovascular diseases (Table G in S1 File) and the other including them (Table H in S1 File).

In the no disease dataset, there were no significant differences observed in TE expression levels between the oldest (]50–70] years, n = 17) and the youngest males ([20–50] years, n = 28) (Wilcoxon test: p-value = 0.371) neither between the oldest females (]50–70] years, n = 14) and the youngest females ([20–50] years, n = 21) (Wilcoxon test: p-value = 0.454). Moreover, we observed no difference in TE expression between the oldest males and the oldest females (Wilcoxon test: p-value = 0.399). We then studied more precisely the effect of age in TE expression using 10-year ranges (age groups: [20–30], ]30–40], ]40–50], ]50–60], and ]60–70]). Again, global expression of TEs was not significantly associated with sex, age in females, or age in males (p-values = 0.788, 0.585, and 0.471, respectively) (Fig 3). Focusing on TE classes/orders/superfamilies (LTR, SVA, SINE, LINE, or DNA elements), active TEs (SVA, HERVK, AluS, L1, or AluY), Y-enriched, Y-depleted, or TEs with no copies in the Y chromosome, we found no significant association between TE expression and both sex and age group. Regarding TE subfamilies, none of them were upregulated in oldest males compared to youngest males or when comparing oldest females with youngest females (panels A and B of Fig O in S1 File). However, we found several TE subfamilies significantly differentially expressed between some age groups, particularly in males (panel B of Fig O, Fig P, and Table I in S1 File). Interestingly, the ]30–40] years old group shows the highest number of significantly differentially expressed TE subfamilies when compared with other age groups in males (panel B of Fig O and

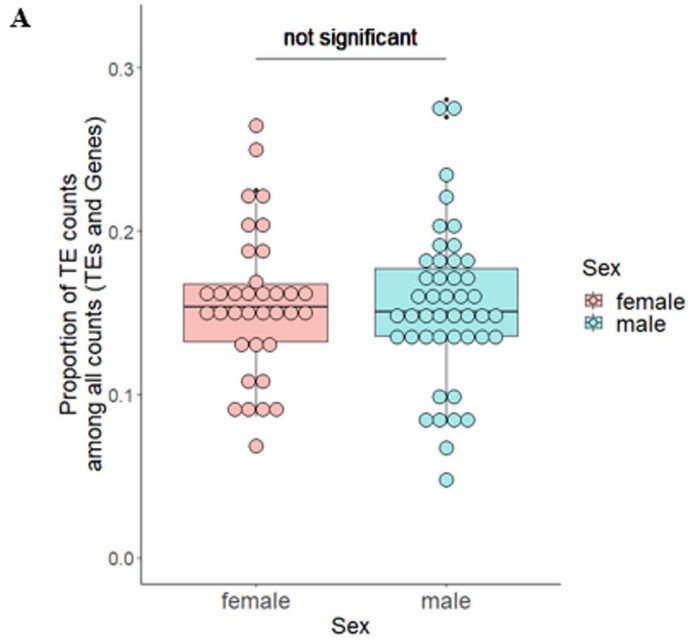

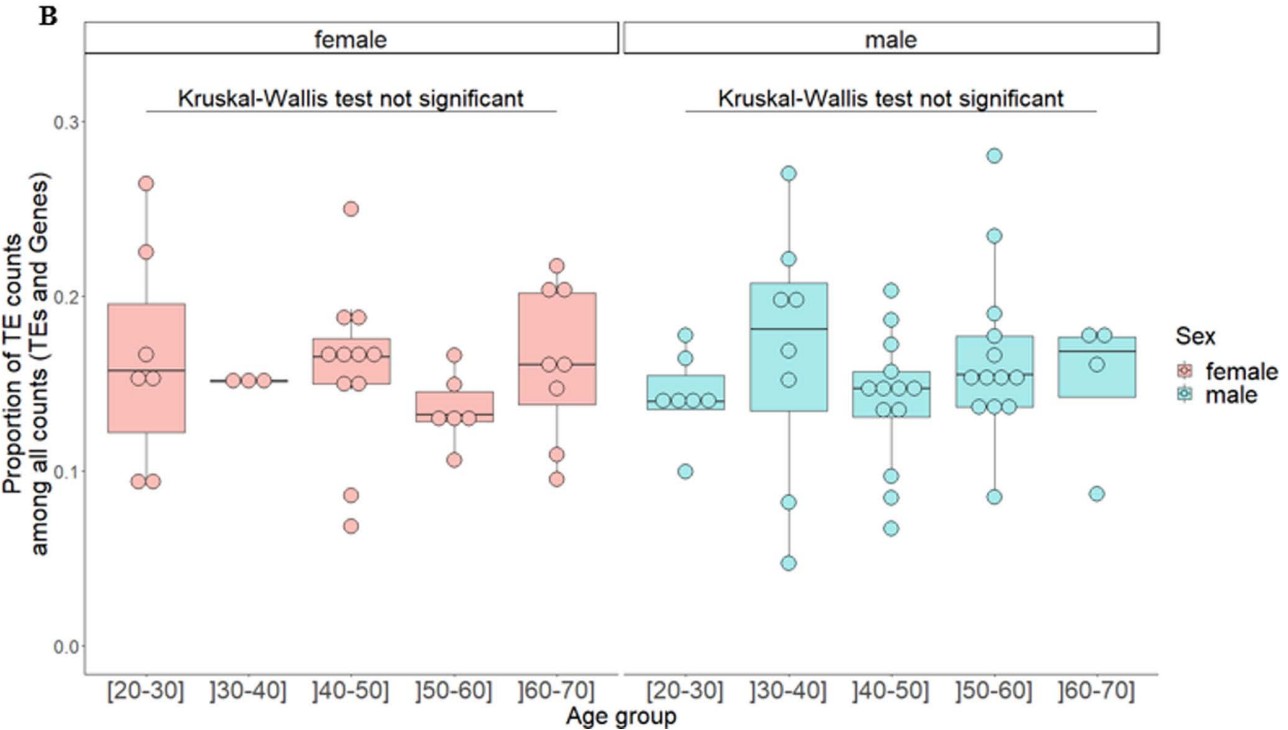

**Fig 3. TE expression in the filtered GTEx dataset (no disease group) according to sex (A) regardless of age group or (B) also according to age group.** Global TE expression was measured in each karyotype (x-axis) as the proportion of TE read counts among all read counts (TEs and genes) (y-axis). In this calculation, read counts were used after DESeq2 normalization ("normalized counts") to remove any depth sequencing bias. Each dot represents one individual. Dots are colored according to sex. Sex variable has two levels: female, male. Age group has five levels: [20-30], ]30-40], ]40-50], ]50-60], and ]60-70]. (A) Proportion of TE read counts among all read counts was modeled using a linear model with sex and age group as independent variables. From this model, the p-value testing the effect of sex on proportion of TE read counts among all read counts adjusted on age group was not significant (p-value = 0.788). (B) A Kruskal-Wallis test comparing proportion of TE read counts among all read counts across age groups was used in females and then in males. This test was not significant in females (p-value = 0.585) as well as in males (p-value = 0.471).

Table I in S1 File). In addition, upon comparing TE expression between males and females in every age group, we did not observe an elevated number of TE subfamilies upregulated in males associated with age (panels C-G of Fig O and Table J in S1 File). Only three TE subfamilies were upregulated in males compared to females adjusted on age group, none were upregulated in females compared to males (Fig 4 and Table J in S1 File). Mosaic Y chromosome loss in white blood cells is a phenomenon whose occurrence increases with age [41]. Considering the potential implication of Y chromosome loss in the absence of difference in TE expression levels between the oldest males and the youngest males, we compared the global expression of the genes carried by the Y chromosome between age groups. No significant difference in the global expression of the genes carried by the Y chromosome between age groups was found (Kruskal-Wallis test: p-value = 0.823) (Fig Q in S1 File) as a proxy of the absence of significant Y chromosome loss.

Including patients who presented or had presented cancers and/or cardiovascular diseases provided similar results regarding overall TE expression and upregulated TE subfamilies (Figs R-T and Tables K-L in S1 File).

Given that filtering the GTEx dataset is likely to diminish the statistical power of our analysis, we repeated the same analysis without filtering based on the individual's ethnicity (large GTEx dataset) (Table M in S1 File). We identified the variable "COHORT" (2 levels: Organ donor (OPO) individuals, Postmortem individuals), referring to the condition in which the organ donation was held, as a potential confounding factor that we have considered in the subsequent analyses. We found no effect of sex on TE expression after adjustment on age group in Organ donor (OPO) individuals nor in Postmortem individuals (panel A of Fig U in S1 File) no relation between age group and TE transcript amounts in the different

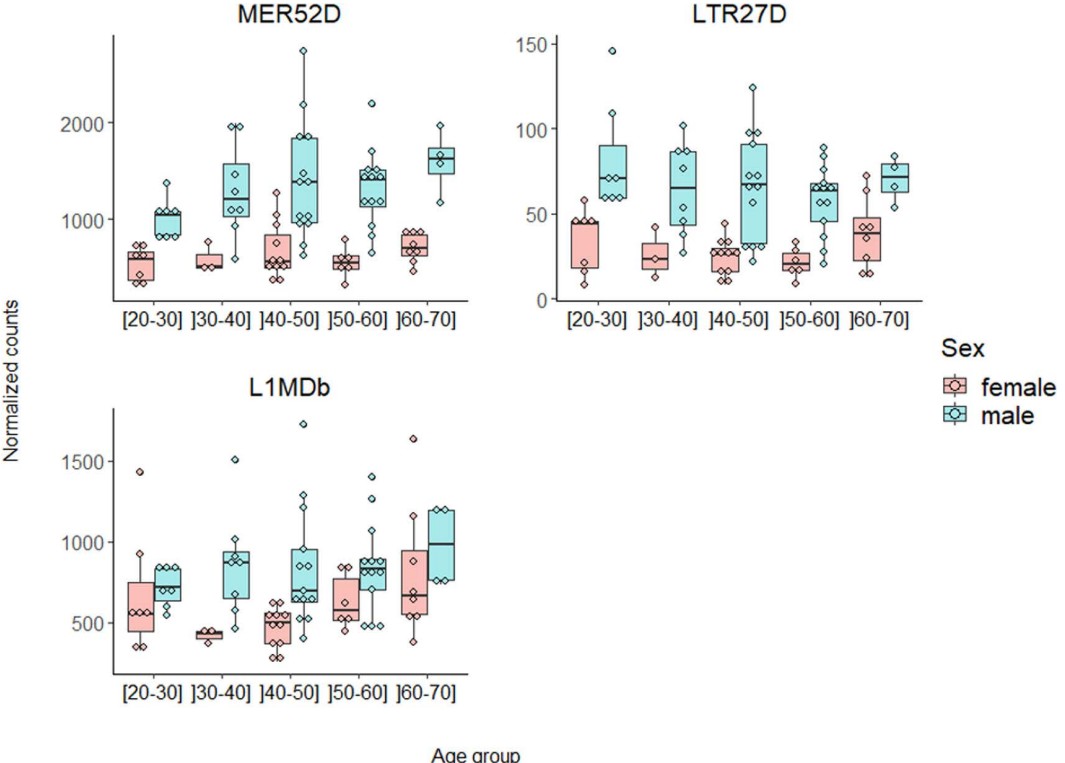

**Fig 4. Normalized counts according to sex and age group for the 3 TE subfamilies that are differentially expressed between males and females adjusted on age group in the filtered GTEx dataset (no disease group).** Specific TE subfamily expression was estimated using the number of read counts after DESeq2 normalization ("normalized counts") to remove any depth sequencing bias. All of these 3 TE subfamilies had at least one copy on the Y chromosome and one copy on the X chromosome, but none are enriched on the Y chromosome. For boxplots of normalized counts according to sex and age group for the significantly differentially expressed TE subfamilies according to age group see Fig P in S1 File.

levels of the variables COHORT (Organ donor (OPO) individuals, Postmortem individuals) and sex (females, males) (panel B of Fig U in S1 File).

Overall, using published transcriptomic data of males and females with a priori normal karyotype (46,XY for males and 46,XX for females), we did not find a clear association between general TE expression and age. Particularly, TE expression in old males compared to old females did not overall increase contrary to our initial hypothesis. Nevertheless, we identified several differentially expressed TE subfamilies between males and females. Additionally, our results showed a higher number of differentially expressed TE subfamilies associated with age in males compared to females.

## Discussion

In this study, we investigated the relationship between TE expression and sex chromosome karyotype composition using transcriptomic data obtained from individuals with different karyotypes (46,XX, 46,XY, 47,XXY and 47,XYY). We observed that sex chromosome content could influence TE expression in humans. The presence of an additional Y chromosome was significantly associated with increased expression of some TE groups (LTR elements, including HERVK elements, and SVA elements). It seemed that the presence of an additional X chromosome tended also to result in an elevated abundance of TE transcripts but to a lesser extent since comparing TE expression between 47,XXY males and 46,XY males almost never reached the significant threshold in any TE group considered. Moreover, contrary to our expectations, analysis of published transcriptomic data from males and females with normal karyotypes revealed no significant increase in TE expression in older males compared to older females.

Our findings suggest a potential association between the presence of the Y chromosome and elevated TE transcripts, aligning with the toxic Y hypothesis. Interestingly, in *D. melanogaster,* the observation of an additional Y chromosome (in XXY females and in XYY males) was associated with an increased deregulation of Y-linked TE insertions [28]. Moreover, the authors also found that TE silencing mechanisms were compromised even in young male flies with a reduction of H3K9me2 histone marks at TE insertions [28]. In crows (*Corvus cornix* and *Corvus corone*), the sex-determination system is ZW (Z being an equivalent of the X chromosome and W an equivalent of the Y chromosome in the XY sex-determination system). In this species, around 10% of the TEs transcribed were differentially expressed between males (ZZ) and females (ZW) in several tissues (liver, spleen, gonad) and TE copies carried by the W chromosome are more highly expressed than copies carried by autosomes and the Z chromosome. Over 95% of sex-differentially expressed TEs were over-expressed in females compared to males, and the majority of these were carried by the W chromosome. Few copies of TEs carried by the autosomes or the Z chromosome were overexpressed in females, suggesting that the W chromosome could influence the expression of TEs on other chromosomes [42] as suggested by Y chromosome data herein. In a study in rats, a low number of TE subfamilies were sex-differentially expressed and were not consistently and predominantly overexpressed in one sex rather than the other in all tissues studied [43].

In humans, differential TE expression according to sex has already been studied using the GTEx dataset, showing that the number of differentially expressed TE subfamilies between males and females varies significantly across different tissues [44]. Nevertheless, in most tissues, including whole blood, differential expression analysis of TE subfamilies between males and females revealed that there was an increased number of upregulated TEs in males compared to downregulated ones [44]. The opposite pattern was observed in few tissues such as breast mammary, muscle skeletal, liver, and thyroid [44]. Hence, our results with whole blood GTEx samples of males and females are in agreement with those reported using also whole blood GTEx samples even though different bioinformatic approaches were used. Differential TE expression in males and females in these different species may be due to the addition of a large number of copies of TEs carried by the Y (or W) chromosome or by the redistribution of heterochromatin marks to silence the numerous repetitive sequences carried by the Y (or W) chromosome [13,32,45]. In the latest human Y chromosome assembly to date (T2T-Y), around 85% of the bases making up the Y chromosome are part of repetitive sequences [17] this is far higher than the rest of the human genome (53.9%) [15,16]. However, the human Y chromosome is not enriched in TE sequences

(31.8%) [17] compared to the rest of the genome (46.0%) [15,16]. Thus, the observed increase in TE expression associated with the presence of an additional Y chromosome, compared to the presence of an additional X chromosome, could be explained by the presence of more transcriptionally active TE copies on the Y chromosome. This difference is likely due to the fact that an additional X chromosome is largely inactivated in humans, while a supplementary Y chromosome remains active [46–48]. Another hypothesis could be that the Y (or W) chromosome induces a redistribution of heterochromatin marks to silence the numerous repetitive sequences carried by the Y (or W) chromosome, named the "sink effect". Sink effect relies on the limited quantity of heterochromatin factors in relation to the amount of repetitive DNA. If the limited heterochromatin factors are used to silence the numerous repetitive sequences carried by the Y (or W chromosome), then the presence of this chromosome could reduce heterochromatin formation in other chromosomes and induce the expression of their usually silent genes and TE copies [32,45,49,50]. In Drosophila, a recent study proposed that Y chromosomes might affect heterochromatin formation in a size-dependent manner, thereby impacting gene silencing in other chromosomes through the titration of heterochromatin factors [32]. However, based on histone methylation data in aneuploid Drosophila, the sink effect was influenced by the amount of repetitive DNA added by the additional sex chromosome and was not a specific pattern of the Y chromosome [45]. To fit our data, the sink effect induced by an additional Y chromosome should be more important than those induced by an additional X chromosome in human. Human data on the sink effect induced by sex chromosomes are scarce and focus on the effect of sex chromosomes on autosomal gene expression [51]. Therefore, the sink effect theory on TE expression remained to be explored in humans. Furthermore, the relationship between TE expression and the Y chromosome may vary from one species to another in relation to the difference in the proportion of the genome represented by the Y chromosome, its richness in TEs and potentially active TEs. For instance, the Y chromosome accounts for over 20% of the genome in Drosophila, whereas it represents around 2% of the human genome [17,45,52]. In *D. melanogaster*, the Y chromosome is richer in retrotransposons than the rest of the genome [53] which is not the case in humans [15–17]. The proportion of TE groups within the genome can also vary from one species to another. For example, SINEs have the highest copy number in humans [15,16], whereas in crows there are more copies of LINEs, LTRs and DNA transposons than SINEs [42].

In several animal species, the association between TE expression and aging has already been studied. It is known that L1 elements are upregulated in aging mouse liver [21] and that several TE families are upregulated in old flies compared to young ones [26,28]. Specifically, LINE elements were found to be activated in the aging fly brain leading to an age-dependent memory impairment and shortened lifespan [26]. In humans, increased expression of DNA transposons, LTR retrotransposons and SINE retrotransposons with age has been observed in human fibroblasts [54]. In the same species, differential TE expression with respect to age was also investigated in the GTEx dataset [44]. Considering all tissues, authors found that there were more TE subfamilies that increased expression (679) compared to the ones that decreased expression (63) with age [44]. However, the number of differentially expressed TE subfamilies is strongly dependent on the tissue considered. Particularly, in whole blood, a higher number of TE subfamilies showed a decreased expression with age compared to those with increased expression [44]. Similar results were also observed for HERV elements in whole blood tissue using the GTEx data [55]. However, the expression of TE subfamilies in relation to the interaction between age and sex has never been studied in this dataset. Hence, we explored the association between TE expression and age according to the sex using the GTEx dataset, hypothesizing a greater activation of TEs with aging in males compared to females. However, we could not find any differences in global TE expression levels between older males and older females, or between the oldest individuals and the youngest ones, contrary to the toxic Y hypothesis. Nevertheless, we suggest that a more homogeneous dataset would be needed to further investigate the relationship between TE expression and age before concluding that Y has no toxic effect on human longevity. First, the inclusion of numerous donors with varying conditions and ethnic backgrounds together with factors such as the absence of longitudinal tracking of individuals or survival bias, might introduce biological noise. Secondly, the many different sequencing runs in GTEx data could contribute to technical noise. This noise in the GTEx data alongside with differences in library

preparation, sequencing read length, and sequencing depth between GTEx datasets and the gonosome aneuploidy dataset (see **Materials and Methods**) may also explain the lack of consistency between the TE subfamilies found differentially expressed in these datasets. Furthermore, in humans, it has been recently shown that there is no correlation between retrotransposon expression and chronological age using different human blood tissues from numerous published datasets [56]. Further research could explore the association between the Y chromosome, age, and TE expression across other human tissues using bulk RNAseq or even in specific cell types with single-cell RNAseq. Such studies would benefit from a larger dataset spanning a wide age range and including various chromosomal aneuploidies, not only involving sex chromosomes but also autosomes, to determine whether variations in TE expression are specifically associated with sex chromosome differences. Additionally, they could take advantage of recent advancements such as the latest human Y chromosome assembly, long-read sequencing technologies, and specialized bioinformatic tools [44,55,57].

Sex gap in longevity is a multifactorial trait and the toxic Y effect is just one of the possible causes underlying this phenomenon. Although we did not explore the longevity aspect of toxic Y effect in our study, our results suggest an association between sex chromosome content and TE expression. In particular, the Y chromosome seems to be associated with a TE expression increase, which may have possible detrimental effects. The primary consequence of TE expression as individuals age is the potential for inducing genomic instability through insertional mutagenesis and/or the generation of insertions or deletions. This is facilitated by the occurrence of double-stranded DNA breaks required for TE reinsertion [58,59]. However, the generation of new somatic mutations with age and the impact of these new TE insertions in the aging process is still to be fully determined in humans [60]. Additionally, the sole expression of TEs can also modify in many different ways the expression and structure of genes putatively causing physiological changes that can in turn reduce longevity [61]. Indeed, a recent study in *D. melanogaster* suggested that the number of TE insertions does not increase with age [62]. Nonetheless, diminishing the expression of two specific TEs, *412* and *roo*, resulted in an extension of longevity [62]. This implies that the expression of TEs, rather than their insertion, play a role in influencing longevity [62]. To emphasize the potential impact of TE expression on aging, a recent study demonstrated correlations between LTR and LINE expression in human blood tissues and biological aging being related with events like cellular senescence, inflammation, and type I interferon response [56]. Also, SINE expression was associated with DNA repair processes, suggesting a potential relationship between SINE expression, DNA damage and genome instability [56]. Indeed, the detrimental effects of TEs can arise through diverse mechanisms including the accumulation of TE transcripts into the cytoplasm, translation of TE transcripts into proteins, and their interaction of TE transcripts with other genes or proteins (such as the inhibition of tumor suppressor protein). These interactions may activate the immune system, contributing to the "inflammaging" process and potentially promoting cancer development [29–31].

This work opens a new window to study the toxic effect of the Y chromosome in human longevity with a particular emphasis on TEs.

## Materials and methods

### Ethics statement

All individuals included in the gonosome aneuploidy datased provided a written consent form for genetic analysis and the study was approved by the ethics committee of the Lyon university hospital (number of the ethics committee: 22_385, number in the register of the *Commission nationale de l'informatique et des libertés*: 22_5385).

### Constitution of the datasets used

**Gonosome aneuploidy dataset: Biological sample collection for 46,XX; 46,XY; 47,XXY; 47,XYY individuals.** We obtained blood samples from 24 individuals: six females (46,XX), six males (46,XY), and 12 males with sex chromosome aneuploidies (eight patients with a 47,XXY karyotype and four with a 47,XYY karyotype). All 47,XXY males and 47,XYY

males had homogenous 47,XXY or 47,XYY karyotype respectively except for one 47,XXY male (95% of mitoses were XXY, 3% XY, and 2% XXXY) and one 47,XYY male (97% of mitoses were XXY and 3% XY). The median age of the individuals was 30 years old (range: 15 – 63).

Blood samples were collected on a PAXgene tube, kept two hours at room temperature, and then frozen at -80°C until extraction. Blood samples were thawed and then RNA was extracted on a Maxwell RSC system (Promega, Wisconsin, USA) with the Maxwell RSC SimplyRNA Blood kit. Quantity (fluorometric concentration) and quality (RNA Integrity Number, RIN) of the extract were assessed by a fluorometric measurement (Quant-iT RNA Assay Kit, Broad Range from ThermoFisher Scientific, Massachusetts, USA) on a Spark reader (Tecan, Männedorf, Switzerland) and by electrophoresis (TapeStation 4200 System from Agilent Technologies, California, USA), respectively. Each sample was of acceptable RNA quantity (fluorometric concentration of nucleic acids between 26.9 and 109.3 ng/µl) and quality (RIN between 5.8 and 9.3).

Libraries were prepared from 300 ng of total RNA using the Illumina TruSeq Stranded Total RNA Lp Gold kit (Illumina Inc, California, USA) (stranded total RNAseq with ribodepletion libraries). Size homogeneity of the fragments was verified using the TapeStation 4200 System. Average length of fragments was between 405 and 498 bp (including adapters). Library concentrations were assessed by fluorometric measurement (Quant-iT 1X dsDNA HS Assay Kit from ThermoFisher) on a Spark reader. Concentrations were between 19.3 and 133.0 nmol/L. Indexed DNA libraries were normalized to 1.5 nmol/L and then pooled in equal volumes before sequencing.

Sequencing was performed on a NovaSeq 6000 system (Illumina Inc, California, USA) in paired-end mode (2 x 151 cycles) with a median of 174 million (Min: 132 – Max: 206 million) pairs of reads per sample (after trimming, see below). Demultiplexing was performed with bcl2fastq (v2.20) to obtain FASTQ files. The 24 individuals were divided into two sequencing runs (run 1: 13 individuals, run 2: 11 individuals) (Table A in S1 File).

**GTEx datasets.** The Genotype-Tissue Expression (GTEx) project regroups transcriptomic data (non-stranded polyA+ libraries, 75x2 paired-end sequencing) from a multitude of post-mortem organ donors and from a multitude of tissues. We focused on whole blood tissue to constitute two datasets: one with selection filters to study the most similar individuals as the gonosome aneuploidy dataset generated in this study (one filtered GTEx dataset excluding individuals with current or antecedents of cancer or cardiovascular diseases and one filtered GTEx dataset including them, see below) and one with fewer selection filters (large GTEx dataset).

Filtered GTEx dataset (no disease group): to constitute this dataset, we extracted raw expression data (FASTQ containing Illumina reads) from the human expression atlas database GTEx via the dbGaP portal (dbGaP accession number phs000424.v8.p2). We considered only data of RNA-seq ("Assay.Type" variable) for which a file with sra extension was available ("DATASTORE.filetype" variable) and generated from RNA extracted from whole blood ("body_site" variable). To reduce data heterogeneity and to get closer to the individuals we sampled in parallel to constitute the gonosome aneuploidy dataset generated herein (see above), we focused on: data collected in non-latino nor hispanic white individuals ("RACE" and "ETHNCTY" variables) and excluded data from individuals with persistent infection by the human immunodeficiency virus (HIV), the hepatitis C virus (HCV) or the hepatitis B virus (HBV) ("LBHCV1NT","LBHBHCVAB", "LBHIV1NT", "LBHIVAB", "LBHIVO", and "LBHBSAG" variables), with dementia ("MHALZDMT" and "MHALZHMR" variables), with current or antecedents of cancer ("MHCANCER5", "MHCANCERC", and "MHCANCERNM" variables), or cardiovascular diseases ("MHHRTATT", "MHHRTDIS", and "MHHRTDISB" variables). At this step, we removed samples identified by the GTEx analysis working group as suboptimal for use in analysis ("SMTORMVE" variable) and we kept only samples for which the material type experiment was described as RNA:Total RNA ("material type exp" variable). Then, we downloaded the remaining FASTQ files (n = 104). For all subsequent analyses, we generated an "age group" variable from the individuals' age by using five classes of 10-year ranges: [20–30], ]30–40], ]40–50], ]50–60], ]60–70]. Then, using PCA on gene expression or TE expression in the whole dataset obtained and in subsets according to sex or by age group, we identified a strong clustering of individuals

who were organ donors OPO (Organ Procurement Organization), who were distinct from individuals who were not (Postmortem individuals) from the "COHORT" variable (Fig V in S1 File). We also found that the "COHORT" variable had a statistically significant relation with the age group (Fisher test: p-value < 0.001) but not with the sex (Fisher test: p-value = 0.185). Therefore, we considered the "COHORT" variable as acting as a confounding factor and we excluded the smallest group of individuals, i.e., individuals who were not organ donors OPO (n = 24). Finally, we obtained 80 files suitable for analysis (Fig N in S1 File) (see Table G in S1 File for repartition of samples according to age and sex). The median number of paired-reads (after trimming, see below) was about 44.6 million (Min: 27.0 – Max: 141.1 million) for the 80 FASTQ.

Filtered GTEx dataset (group including individuals with current or antecedents of cancer or cardiovascular diseases): we constituted the same cohort of individuals as before but including 22 individuals with current or antecedents of cancer or cardiovascular diseases (Table H in S1 File). Therefore, this dataset was composed of 102 individuals. The median number of paired-reads (after trimming, see below) was about 44.3 million (Min: 27.0 – Max: 141.1 million) for the 102 FASTQ.

Large GTEx dataset: to constitute this dataset, we used the 318 curated files containing raw human blood expression data (FASTQ containing Illumina reads) of the Bgee project (see supplementary data of [63]) originating from dbGaP (dbGaP accession number phs000424.v6.p1). We excluded one male suspected of having sex chromosome aneuploidy (such as 47,XXY) because of the expression of *XIST* like individuals in possession of two X chromosomes and the expression of *USP9Y* like individuals in possession of one Y chromosome. We generated an "age group" variable from the individuals' age by using five classes of 10-year ranges:[20–30], ]30–40], ]40–50], ]50–60], ]60–70] and we found a significant statistical relation between sex and age group variables (Pearson's Chi-squared test: p-value = 0.013) (Table M in S1 File). Again, using PCA on gene expression or TE expression, we identified a strong clustering of individuals who were organ donors OPO (Organ Procurement Organization), who were distinct from individuals who were not (Post-mortem and Surgical individuals) from the "COHORT" variable. We also found that the "COHORT" variable had a significant statistical relation with the age group (Pearson's Chi-squared test: p-value < 0.001, excluding the 3 individuals with "COHORT" = Surgical) and with the sex group (Pearson's Chi-squared test: p-value = 0.017, excluding the 3 individuals with "COHORT" = Surgical). Then we considered the variable "COHORT" as a potential confounding factor and added it in the DESeq2 statistical model (see below). Therefore, we excluded 3 more individuals for whom "COHORT = Surgical" to guarantee a sufficient number of individuals in each COHORT, sex, and age group levels. Thus, 314 files were used in downstream analysis.

## Bioinformatic analysis

See pipeline in Fig A in S1 File.

**Input files.** FASTQ files were used after a quality control step with FastQC (v0.11.9) and Trimmomatic (v0.33) softwares [64]. The whole human reference transcriptome (Human Release 32 GRCh38/hg38, gencode v32) was downloaded from the University of California Santa Cruz (UCSC) table browser (https://genome.ucsc.edu/cgi-bin/hgTables?hgsid=814613547_qUmIEGan4K2f39cHvb9AgrKdRVdK&clade=mammal&org=Human&db=hg38&hgta_group=genes&hgta_track=knownGene&hgta_table=0&hgta_regionType=genome&position=chr1%3A11%2C102%2C837-11%2C267%2C747&hgta_outputType=primaryTable&hgta_outFileName=) with output format = "sequence" (fasta file) and used for the differential gene expression analysis. The downloaded file contained 247,541 transcript sequences (5' and 3' untranslated regions and coding DNA sequences of each gene were included, introns were discarded, and repeated regions were masked). The GTF file corresponding to release 32 GRCh38/hg38 of the human reference genome was downloaded from the gencodegenes website https://www.gencodegenes.org/human/ (content = "comprehensive gene annotation", regions = "CHR"). This file provided a correspondence between transcripts (227,463 transcript names) and genes (60,609 genes, including 19,947 protein coding genes). The human reference sequence file for repeats (including

TE insertion sequences) was downloaded as a fasta from the UCSC table browser (https://genome.ucsc.edu/cgi-bin/hgTables?hgsid=791366369_Zf0cNT7ykVM0ZQ0zErZRArSgvMEO&clade=mammal&org=Human&db=hg38&hgta_group=allTracks&hgta_track=rmsk&hgta_table=0&hgta_regionType=genome&position=chr1%3A11%2C102%2C837-11%2C267%2C747&hgta_outputType=primaryTable&hgta_outFileName=) including the RepeatMasker track from the "Dec. 2013 GRCh38/hg38" assembly of the human reference genome. This file contains sequences of 4,886,205 insertion copies, after excluding transfer RNAs, ribosomal RNAs, small RNAs, repeats U1 to U13, and satellites (microsatellites, GSATX, HSAT5). From this file, we built the "rosetta" file (available here: https://dx.doi.org/10.6084/m9.figshare.28365464), which provided a correspondence between the 4,886,205 insertion sequences and insertion subfamily names (1,270 repeat subfamilies). Since we did not have access to the genome of the participants, we could not determine the location of each repeat copy in each individual, which may vary from one to another. Therefore, the rosetta file allowed us to regroup read counts per repeat copy into read counts per repeat subfamily, independently of their location in the genome.

**Alignment and read counting.** Gene expression was quantified with Kallisto software (v0.46.1) [65] using the transcript sequences from the whole human reference transcriptome. The --rf-stranded option was added to deal with reverse stranded reads when analyzing RNAseq data of the gonosome aneuploidy dataset (contrary to GTEx datasets as GTEx data are non-stranded). Alignments to the reference transcriptome resulted in high mapping efficiency (Table A in S1 File). Repeat expression (alignment and read count) was analyzed with the TEcount (v1.0.0) software from TEtools [36]. This tool uses Bowtie2 [66] as aligner, and we added the --nofw option to deal with reverse stranded RNAseq data when analyzing RNAseq data of the gonosome aneuploidy dataset (contrary to GTEx datasets).

## Statistical analysis

Statistical analysis was performed using the R software (v3.6.3 to 4.3.2). The significance threshold was set at 0.05 for all statistical tests performed. The GTEx datasets and the gonosome aneuploidy dataset were computed separately using the same procedure described below. We considered as gene or TE "expression" the number of reads aligned (i.e., counts) on each reference transcript or TE sequence and as "differential expression analysis" the statistical procedure to compare gene or TE expression between several conditions (i.e., sex, age group, or karyotype).

**Data preparation.** Differential expression analysis was performed separately for genes and for TEs. Two files were prepared for each purpose. Since one gene can have several transcripts, the tximport R package (v1.14.2, Bioconductor) was used to sum the read counts per gene transcript into read counts per gene using the GTF file mentioned above. Then, a txi file containing the gene identifier (Ensembl nomenclature), the average gene length, and the estimated counts for all genes per sample was obtained. For DE analysis on TEs, we normalized the repeat subfamily read counts based on the normalization factor previously calculated using gene read counts to avoid masking relevant biological information [36]. To estimate the normalization factor, the gene identifiers and read counts per gene were extracted from the txi file and merged with the TEcount file containing the repeat subfamily names and read counts per repeat subfamily. The length could not be included in the merged table since this variable was not available from the TEcount table (each repeat subfamily is a collection of copies displaying a diversity of lengths, it is thus difficult to estimate the length of a TE subfamily). This merged table was then imported into the DESeq2 tool (R package DESeq2 v1.26.0, Bioconductor [67,68], using the DESeqDataSetFromMatrix function, which normalized the read counts so as to remove any depth sequencing bias. From this step on, we removed read counts on the following 24 snRNA, snpRNA, scRNA, or remaining satellites and rRNA repeat subfamilies: 7SLRNA, 7SK, LSAU, D20S16, REP522, SATR1, SATR2, ACRO1, ALR/Alpha, BSR/Beta, CER, 6kbHsap, TAR1, SST1, MSR1, SAR, GA-rich, G-rich, A-rich, HY1, HY3, HY4, HY5,5S. This step allowed us to only consider the 1,246 subfamilies corresponding to TE subfamilies out of the 1,270 repeat subfamilies contained in the rosetta file. Then, genes and TE subfamilies with zero read counts in all individuals were removed leaving 1,212 TE subfamilies to study.

For the DE analysis on genes, as the length was available for genes, a more reliable procedure was performed. The raw txi file was directly imported in DESeq2 (DESeqDataSetFromTximport function) and the average length of each gene

could be used as a correction term (offset) in the DESeq2 statistical model described below to consider the length bias [67,68]. This procedure normalized the read counts in order to remove any depth sequencing bias and also bias related to sequence length.

**Comparison of the global expression of TE between sex, age group or karyotypes.** We first looked at the global TE expression rather than the TE subfamily expression separately. To do that, using the merge table imported into DESeq2, we summed all the normalized counts of each TE subfamily per individual and then we divided by the total number of normalized counts on genes and TE subfamilies per individual. A pairwise Wilcoxon test was used to compare global TE expression between karyotypes in the gonosome aneuploidy dataset. In GTEx datasets, linear models or Wilcoxon tests were used, as appropriate, to compare global TE expression between females and males (adjusted on age group and COHORT variables when necessary), and between the youngest individuals ([20–50] years, obtained by combining individuals from [20–30],]30–40], and ]40–50] age groups) and the oldest individuals (]50–70] years, obtained by combining individuals from ]50–60], and ]60–70] age groups). Kruskall-wallis tests or ANOVA were also used, as appropriate, to test the relation between global TE expression and age group in the GTEx datasets in males and females separately. The same procedure was made to looked at expression of TE classes/orders/superfamilies (LTR-retrotransposons, non-LTR retrotransposons: SVA, non-LTR retrotransposons: SINE, non-LTR retrotransposons: LINE, DNA transposons) or some TE groups (SVA, HERVK, AluS, L1, AluY) by summing only normalized counts of TE subfamilies belonging to this TE class/order/superfamily or this TE group according to the "class/family" column of the hg38.fa.out.gz file downloaded from https://www.repeatmasker.org/species/hg.html (track hg38 RepeatMasker open-4.0.6 - Dfam 2.0) or according to the name of TE subfamilies when necessary.

**Differential expression analysis on genes and TE subfamilies.** Then, we performed a differential expression analysis for each gene and each TE subfamily separately.

The R package DESeq2 (v1.26.0, Bioconductor) [67,68] was used to fit a negative binomial model on the log2 of the normalized counts (dependent variable) according to two independent variables: sex (factor with 2 levels: female, male) and age group (factor with 5 levels: [20,30], ]30,40], ]40,50], ]50,60], ]60,70]) and the interaction of the two for the GTEx datasets, or karyotype (factor with 4 levels: XX, XY, XXY, XYY) and sequencing batch (factor with 2 levels: first batch, second batch) for the gonosome aneuploidy dataset. The statistical model provided variation of expression of a gene or a TE subfamily between two conditions expressed as log2 fold-change (log2FC) (i.e., log base 2 of the ratio of predicted normalized counts between the two conditions). Log2FC were compared to 0 with a Wald test. A likelihood-ratio test (LR test) was also performed to assess whether adding an independent variable in the model provided relevant information compared to the null model, i.e., to test whether gene or TE subfamily expression was globally impacted by an independent variable having more than two levels (age group or karyotype variables). A 0.05 False-Discovery Rate (FDR) threshold value was used for significance and p-values were adjusted for multiple-testing using the Benjamini & Hochberg procedure [69]. The thresholds for the outlier and the low mean normalized read count filters were set as default. Graphics were generated after the data transformations proposed by the DESeq2 package authors: variance stabilizing transformation for PCA, adaptive shrinkage for Minus-Average plots (MA plots), log2 transformation of the normalized counts for heatmaps [67,70]. When specified, the command removeBatchEffect function (for unbalanced batches) from the limma R package (v3.46.0) was used before PCA representations, heatmap representations and before building the different boxplots of proportion of TE read counts or normalized counts to deal with batch effect in the gonosome aneuploidy dataset or COHORT effect in the large GTEx dataset. When necessary, gene name, gene description, and gene localization were determined using the GTF file and the command gconvert from the R package gprofiler2 (v0.1.9) [71,72] from the Ensembl gene (ENSG) identifier.

**Overlap between the DEGs generated in our study and in published studies.** The DEGs generated in our study in the 47,XXY karyotype compared to the 46,XY karyotype was compared to those found in the supplementary materials of Zhang et al. study [73] and to confirmed XCI-escapees in supplementary materials of Wainer-Katsir et al. study [74].

**GO term enrichment analysis.** A gene ontology (GO) analysis was performed using the lists of significantly upregulated genes (adjusted p-value < 0.05 and log2-fold change > 0) and significantly downregulated genes (adjusted p-value < 0.05 and log2-fold change < 0) obtained for each of the six pairwise karyotype comparisons (47,XYY vs 46,XX, 47,XXY, 46,XX, 46,XY vs 46,XX, 47,XYY vs 46,XY, 47,XXY vs 46,XY, 47,XYY vs 47,XXY) after adjustment on the batch effect. To generate lists of significant GO terms (S3 Data), we used the gost function from the R package gprofiler2 (v0.1.9) [71,72] with the g:SCS algorithm for multiple testing correction and a significance threshold set at 0.05. Other parameters were set to default. To generate plots (Figs E-G in S1 File), we used the compareCluster function from the R package clusterProfiler (v4.10.0) [75] with the Benjamini & Hochberg procedure [69] for multiple testing correction and a significance threshold set at 0.05. Other parameters were set to default. We generated a plot concerning terms in the GO database, KEGG database and Reactome database. Then, we plotted specifically the significant terms from the Reactome database including the word "senescence" in their description (Fig H in S1 File).

**Y enrichment of TE copies.** For all of the 1,246 TE subfamilies, an exact Binomial test was performed to test if the observed proportion of TE copies for one TE subfamily carried by the Y chromosome was equal to the expected proportion of 1.85%. The expected proportion of 1.85% (57200/3088200) was calculated according to the length of Y chromosome (57200 kb) compared to the sum of the length of all chromosomes (3088200 kb) [35]. The observed proportion of TE copies corresponded to the ratio between the observed number of TE copies from a TE subfamily carried by the Y chromosome and the observed total number of TE copies from the same TE subfamily in the human reference sequence file including TE insertion sequences (see above). The expected number of TE copies from a TE subfamily carried by the Y chromosome was determined as the total number of TE copies multiplied by 1.85%. The exact Binomial test was two-sided. A 0.05 False-Discovery Rate (FDR) threshold value was used for significance and p-values were adjusted for multiple-testing using the Benjamini & Hochberg procedure. S6 Data regroups the observed and expected numbers of TE copy carried by the Y or the X chromosome and the adjusted p-values of the exact Binomial test obtained for each of the 1246 TE subfamilies. Then, using S6 Data, we made 6 groups of TE subfamilies: one concerning Y-enriched TE subfamilies (binomial test adjusted p-value < 0.05 and number of observed copies located on Y chromosome > expected), one concerning Y-depleted TE subfamilies (binomial test adjusted p-value < 0.05 and number of observed copies located on Y chromosome < expected), one concerning neither Y-enriched nor Y-depleted TE subfamilies (TE subfamilies that did not meet the conditions for Y-enriched nor Y-depleted TE subfamilies), one concerning TE subfamilies with no copies carried by the Y chromosome, one concerning TE subfamilies with no copies carried by the X chromosome, and one concerning TE subfamilies with no copies carried by either the Y or X chromosomes. Finally, a pairwise Wilcoxon test was used to compare TE expression between karyotypes in these 6 TE groups in the gonosome aneuploidy dataset. In GTEx datasets, linear model was used to compare TE expression between females and males, adjusted on age group, and kruskall-wallis test was used to test the relation between TE expression and age group in males and females separately.

**Overlap between 3 kb upstream regions of differentially expressed genes and differentially expressed TE subfamilies.** First, we downloaded a bed file for repeats from https://hgdownload.soe.ucsc.edu/goldenPath/hg38/bigZips/ (hg38.fa.out.gz file, last update 2014-01-15 20:56) and we generated a bed file for genes from the GTF file Release 32 GRCh38/hg38 previously downloaded. From the gene bed file, we generated a bed file containing the 3 kb upstream region from the transcription start site of all genes. Then, we filtered this bed file using the list of differentially upregulated (downregulated, respectively) genes obtained for each pairwise karyotype comparison. We filtered the repeat bed file using the list of differentially upregulated (downregulated, respectively) TE subfamilies obtained for each pairwise karyotype comparison. Then, we used the R package RegioneR v1.34.0 [76] (toGRanges and numOverlaps functions) to obtain the number of TE copies whose sequence overlaps the 3 kb upstream sequence of any gene using the filtered bed files previously created. A permutation test was performed to test whether the frequency of genes with TE copies overlapping their 3 kb upstream region was significantly greater or lower than expected by chance. We used the permTest

function with the following options:ntimes = 5000, randomize.function = resampleRegions, evaluate.function = numOverlaps, count.once = TRUE, alternative = "auto", and the 3 kb upstream region of all genes as universe. We made this procedure four times for each pairwise karyotype comparisons: one considering TE copies from upregulated TE subfamilies and upregulated genes, one considering TE copies from upregulated TE subfamilies and downregulated genes, one considering TE copies from downregulated TE subfamilies and upregulated genes, and one considering TE copies from downregulated TE subfamilies and downregulated genes.

**Overlap between intragenic regions of differentially expressed genes and differentially expressed TE subfamilies.** We also looked for overlap between TE copy sequences of upregulated TE subfamilies and sequences of upregulated genes using bedtools intersect (v2.30.0) with -s option to consider the strandness. Then, we made the same procedure considering TE copy sequences of downregulated TE subfamilies and sequences of downregulated genes. Then, we performed an exact binomial test for each pairwise karyotype comparison which compared the observed proportion of upregulated genes containing at least one TE copy from one of the upregulated TE subfamilies among all the upregulated genes to the expected proportion of genes containing at least one TE copy from one of the upregulated TE subfamilies among all genes. The same procedure was also applied considering then downregulated genes and downregulated TE subfamilies.

## Supporting information

**S1 File. Supplementary materials include supplementary text [77], supplementary figures and supplementary tables.**
(PDF)

**S1 Data. List of all differentially expressed genes on X chromosome or Y chromosome according to karyotype using the likelihood-ratio test in the gonosome aneuploidy dataset, first considering all genes and then considering protein-coding genes only.**
(ZIP)

**S2 Data. List of all differentially expressed genes (regardless if they are protein-coding or not) in each pairwise karyotype comparison after batch adjustment in the gonosome aneuploidy dataset.**
(ZIP)

**S3 Data. GO analysis in the gonosome aneuploidy dataset.**
(ZIP)

**S4 Data. Classification of the differentially expressed TE subfamilies in each pairwise karyotype comparison after batch adjustment in the gonosome aneuploidy dataset.**
(ZIP)

**S5 Data. Number of TE subfamilies upregulated and downregulated per TE family or TE superfamily/order/class in all the pairwise karyotype comparisons.**
(XLSX)

**S6 Data. Y chromosome enrichment test for TE subfamilies.**
(CSV)

**S7 Data. Overlap between genomic positions of upregulated (downregulated, respectively) genes and positions of upregulated (downregulated, respectively) TEs for each pairwise karyotype comparison (when applicable).**
(ZIP)

## Acknowledgments

We thank Yasmine Zerdoumi and Maxime Vallée for their help in the data generation and data analysis, and Véréna Landel (DRS, Hospices Civils de Lyon) for reading the manuscript. We thank Marc Robinson-Rechavi for access to the Bgee database and useful comments on the GTEx analysis and on the manuscript. Part of this work was performed using the computing facilities of the CC LBBE/PRABI-AMSB. Filtered GTEx datasets were generated as part of dbGaP Project #28282: "Studying transposable elements activity in human tissues". We gratefully thank the GTEx group for the data provided.

A preprint version of this article has been peer-reviewed and recommended by PCI Genomics (https://doi.org/10.24072/pci.genomics.100293).

## Author contributions

**Conceptualization:** Cristina Vieira, Gabriel AB Marais, Ingrid Plotton.

**Data curation:** Jordan Teoli, Marie Fablet, Anamaria Necsulea.

**Formal analysis:** Jordan Teoli, Daniel Siqueira-de-Oliveira.

**Funding acquisition:** François Gueyffier, Damien Sanlaville, Cristina Vieira, Gabriel AB Marais, Ingrid Plotton.

**Investigation:** Jordan Teoli, Anamaria Necsulea, Audrey Labalme, Hervé Lejeune, Damien Sanlaville, Claire Bardel, Gabriel AB Marais, Ingrid Plotton.

**Methodology:** Marie Fablet, Damien Sanlaville, Cristina Vieira, Gabriel AB Marais, Ingrid Plotton.

**Project administration:** Cristina Vieira, Gabriel AB Marais, Ingrid Plotton.

**Resources:** Jordan Teoli, Marie Fablet, Anamaria Necsulea, Alessandro Brandulas-Cammarata, Audrey Labalme, Hervé Lejeune, Damien Sanlaville, Claire Bardel, Gabriel AB Marais, Ingrid Plotton.

**Software:** Jordan Teoli, Marie Fablet, Anamaria Necsulea, Daniel Siqueira-de-Oliveira, Alessandro Brandulas-Cammarata, Claire Bardel.

**Supervision:** Cristina Vieira, Gabriel AB Marais, Ingrid Plotton.

**Visualization:** Jordan Teoli, Miriam Merenciano, Cristina Vieira, Gabriel AB Marais, Ingrid Plotton.

**Writing – original draft:** Jordan Teoli, Gabriel AB Marais.

**Writing – review & editing:** Jordan Teoli, Miriam Merenciano, Marie Fablet, Anamaria Necsulea, Daniel Siqueira-de-Oliveira, Alessandro Brandulas-Cammarata, Audrey Labalme, Hervé Lejeune, Jean-François Lemaitre, François Gueyffier, Damien Sanlaville, Claire Bardel, Cristina Vieira, Gabriel AB Marais, Ingrid Plotton.

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
