## [Decision Letter · Decision Letter 0]

Dear Dr Teoli,

Thank you very much for submitting your Research Article entitled 'Transposable element expression with variation in sex chromosome number: insights into a toxic Y effect on human longevity' to PLOS Genetics.

The manuscript was fully evaluated at the editorial level and by independent peer reviewers. The reviewers appreciated the attention to an important problem, but raised some substantial concerns about the current manuscript. Based on the reviews, we will not be able to accept this version of the manuscript, but we would be willing to review a much-revised version. We cannot, of course, promise publication at that time.

If you decide to revise the manuscript for further consideration at PLOS Genetics, please aim to resubmit within the next 60 days, unless it will take extra time to address the concerns of the reviewers, in which case we would appreciate an expected resubmission date by email to plosgenetics@plos.org.

If present, accompanying reviewer attachments are included with this email; please notify the journal office if any appear to be missing. They will also be available for download from the link below. You can use this link to log into the system when you are ready to submit a revised version, having first consulted our Submission Checklist .

PLOS has incorporated Similarity Check , powered by iThenticate, into its journal-wide submission system in order to screen submitted content for originality before publication. Each PLOS journal undertakes screening on a proportion of submitted articles. You will be contacted if needed following the screening process.

To resubmit, log into your Editorial Manager account and select the option 'Revise Submission' in the 'Submissions Needing Revision' folder.

We are sorry that we cannot be more positive about your manuscript at this stage. Please do not hesitate to contact us if you have any concerns or questions.

Yours sincerely,

Edward Chuong

Academic Editor

PLOS Genetics

Giorgio Sirugo

Section Editor

PLOS Genetics

All of the reviewers were enthusiastic about the study overall, but each had multiple concerns with the analyses and inclusion of appropriate controls and statistical comparisons. These will need to be addressed in a revised submission.

Reviewer's Responses to Questions

**Comments to the Authors:**

Reviewer #1: In this manuscript, Teoli et al set out to test whether Y-linked TEs may contribute to the sex gap in longevity in humans, a phenomenon known as the “toxic Y” effect. They analyze two RNA-seq datasets to address this question: (1) expression profiles from blood of groups of patients with different sex chromosome complements (XX, XY, XXY, XYY) and (2) GTEx expression profiles from blood of XX females and XY males across different age groups. In the first analysis, they find a fairly consistent trend where a variety of TE subfamilies show increasing expression across the karyotype gradient XX, XY, XXY, and XYY. In the second analysis they find no difference in TE expression between XX and XY groups or between different age groups within sexes. Unfortunately, after reading the manuscript, I am unsure of what can be concluded from the work presented here. The results from the first analysis are interesting but aren’t directly related to the toxic Y/longevity hypothesis: the upregulation of TEs also occurs in XXY individuals and there is no connection to ageing in this analysis. On the other hand, although the second analysis does test for differences in TE expression across age groups, no significant differences are found.

This manuscript has already been reviewed through PCI Genomics, which I am not familiar with. I largely agree with the previous reviewers comments and the authors revisions, however I have additional feedback that hopefully may be useful.

(1) The authors frame their work as addressing whether Y-linked TE expression could contribute to differences in longevity between sexes but I don’t think that these are the best datasets to test such a question. The GTEx analysis seems to be a negative result but there are so few samples in each age group it is unclear whether the lack of a difference in TE expression with age is real or just a result of limited power. I appreciate the authors acknowledgement of the limitations of their datasets, but I would encourage them to consider whether the toxic Y hypothesis is the best way to frame this work, especially because there also seems to be an effect on TE expression from having an extra X chromosome. Why not just frame this as a test of the effect of sex chromosome complement on TE expression?

(2) The authors briefly mention another important genome-wide effect caused by the Y chromosome, which is that there is evidence that the Y can act as a “sink” for heterochromatin, resulting in the redistribution of heterochromatic marks from the autosomes onto the Y. If this is the case, autosomal genes that are normally silenced or lowly expressed in XX individuals should increase in expression along with Y chromosome dosage (and also potentially with an extra X chromosome as well depending upon whether the extra X is inactivated). It should be possible to test for this effect using the RNA-seq data from XX, XY, XXY, and XYY individuals.

(3) It is striking to me how consistent the pattern of TE upregulation is for the same individuals across multiple TE subfamilies. For example, in Fig 1, the 26 year-old XY male is an outlier in every panel. This could be real biology or an issue with normalization. There is no need to use DESeq normalization before the “Proportion of TE counts” expression calculation used by the authors. Their proportion measurement already accounts for differences in sequencing depth because it is a proportion, rather than simply a count of raw reads. I would encourage the authors to confirm that the patterns in Figures 1 & 2 are reproducible when calculating the proportion measurement from raw counts rather than DESeq normalized counts.

Minor comments:

Line 58: “increased TE activity”: the authors should be careful to make clear that their results are only with respect to TE expression and not activity since they did not search for new TE insertions in their subjects.

The authors calculate the enrichment of various TE subfamilies on the Y chromosome, presumably using the hg38 genome assembly. They may want to consider redoing these calculations using the newer T2T human Y chromosome assembly, which likely contains many more TE copies that were missing from the version in hg38.

The “proportion of TE counts” is a fairly conservative measure, especially when used for all TEs combined since the upregulation of some TE subfamilies will be masked by downregulation of other subfamilies. A paired Wilcoxon test would be much more sensitive if the goal is to test for an overall upregulation of a group of TE subfamilies. This would involve using paired expression values between karyotypes, for example, normalized counts of each TE subfamily from the XX group paired with their normalized counts from the XY group. The test calculates whether the median of the differences between the paired expression measurements differs significantly from zero.

Reviewer #2: review uploaded as attachment (pdf)

Reviewer #3: The manuscript "Transposable element expression with variation in sex chromosome number: insights into a toxic Y effect on human longevity” explores what the role of TE expression might be in sex differences in aging.

This is an interesting unsolved problem in aging research: why do males and females age at different rates (or more specifically, why is lifespan different across sexes)? One intriguing possibility is that the sex chromosomes themselves harbor TEs that, when expressed during age epigenetic/transcriptional dysregulation, may be deleterious—with preferential effects on one sex. There certainly seems to be ample evidence of this in flies, which the authors hope to build upon.

I’m broadly enthusiastic about this manuscript, I think it will be an important finding for the field of TE and aging, and hope that the authors can improve it to the (admittedly, very high) standard of PLoS Genetics.

Here are my major concerns:

1. The authors mention that the human Y chromosome is enriched for TE content compared to other chromosomes, and that this may provide a conceptual basis for male-specific toxic TE effect. However, I would like to point out that the human X chromosome is also extremely enriched for LINE L1 retrotransposons (Bailey, et al. PNAS 2000). The X chromosome is a large chromosome, with 1000 genes. Based on these other arguments, it might seem like females (46 XX, 47XXX) or 47XXY males might have a toxic X effect—although admittedly, only one X should be active in 46XX, 47XXX, or 47XXY individuals if they have stable X chromosome inactivation. Additionally, stating that the Y chromosome is “enriched for TE” should be backed up with some analysis of the age of TE on the Y or the intactness (ability to be transcribed etc)—since not just the presence of TE on the Y, but their ability to be made into RNA through transcription is the mechanistic basis of the toxic Y model. To be fair to the reader, the authors should discuss this alternative possibility (X chromosome/TE toxicity) and explain if they think this is not a viable hypothesis.

2. Mosaic Y chromosome loss in white blood cells is estimated to occur in 20% of men (Thompson, et al. Nature Genetics 2019). Is there any way that the authors can ensure that the 46XY, 47XYY, or 47XXY males do not have mosaic Y loss? I would imagine that this phenomenon would complicate their study.

3. A major concern for this work is that the RNA-seq that is the foundation of the conclusions is based on bulk RNA expression from white blood cells. If there are sex-specific differences in the type of white blood cells, and each type of cell has its own transcriptional profile, and each TE family can respond to cell-type specific transcription factors, then the TE expression differences they see with age/sex are not a reflection of the toxic Y per se, but rather from different cell compositions of blood in age/sex variable manner. There is published work showing that white blood cell composition is slightly different in males and females (Bongen, et al. Cell Reports 2019). A nice piece of data that argues against this concern is that the authors find that the TE families that are upregulated in aged Y chromosome containing individuals are found as TEs enriched on the Y chromosome. This specific concern could be addressed by mining the publicly available scRNAseq datasets from human white blood cells. If we can now compare specific cell types vs specific cell type using the scRNAseq data, can we pinpoint if there is a single cell type that contributes this TE upregulation in Y-chromosome containing individuals? Or do all cells in the blood show similar TE dysregulation? This type of analysis is likely outside of the scope of the current work, but I would encourage the authors to consider utilizing these publicly available data for follow-up studies. They could make a substantial contribution in this area using their analysis pipeline with scRNAseq datasets.

**Have all data underlying the figures and results presented in the manuscript been provided?**

Reviewer #1: None

Reviewer #2: Yes

Reviewer #3: Yes

PLOS authors have the option to publish the peer review history of their article (what does this mean? ). If published, this will include your full peer review and any attached files.

**Do you want your identity to be public for this peer review?** For information about this choice, including consent withdrawal, please see our Privacy Policy .

Reviewer #1: No

Reviewer #2: No

Reviewer #3: No

---

## [Decision Letter · Decision Letter 1]

PGENETICS-D-24-00837R1

Transposable element expression is associated with sex chromosome number in humans

PLOS Genetics

Dear Dr. Teoli,

Thank you for submitting your manuscript to PLOS Genetics. After careful consideration, we feel that it has merit but does not fully meet PLOS Genetics's publication criteria as it currently stands. Therefore, we invite you to submit a revised version of the manuscript that addresses the points raised during the review process.

Please submit your revised manuscript within 30 days Apr 16 2025 11:59PM. If you will need more time than this to complete your revisions, please reply to this message or contact the journal office at plosgenetics@plos.org. Please include the following items when submitting your revised manuscript:

We look forward to receiving your revised manuscript.

Kind regards,

Giorgio Sirugo

Section Editor

PLOS Genetics

Aimée Dudley

Editor-in-Chief

PLOS Genetics

Anne Goriely

Editor-in-Chief

PLOS Genetics

**Additional Editor Comments :**

Please address comments by Reviewer 2.

**Journal Requirements:**

1) Thank you for stating "Fastq files of the gonosome aneuploidy dataset generated in this study were submitted to the European Genome-phenome Archive (EGA ID of the study: EGAS00001007462). " Please note that, though access restrictions are acceptable now, your entire minimal dataset will need to be made freely accessible if your manuscript is accepted for publication. This policy applies to all data except where public deposition would breach compliance with the protocol approved by your research ethics board. If you are unable to adhere to our open data policy, please kindly revise your statement to explain your reasoning and we will seek the editor's input on an exemption."

**Reviewers' comments:**

Reviewer's Responses to Questions

Reviewer #1: The authors have done a nice job addressing my previous comments, I have no further suggestions.

Reviewer #2: Overall, the authors have done an excellent job responding to my concerns, both in their response letter and in their revisions to the manuscript. In particular, the authors did a good job of stating the primary comparisons of interest, and reporting results for these comparisons in a consistent manner throughout the manuscript. In addition, “Data S5” is a very helpful addition (particularly columns E and F), which makes it much easier to see results for TE subfamilies across all comparisons. The clarity and accuracy of the writing has also improved. There are many supplemental data figures and tables contained within the manuscript, which can be difficult to navigate and distinguish between, but these data are well organized and well described and should be a useful resource to other researchers in the field. The central finding that sex chromosomes, particularly the Y chromosome, might be associated with TE expression does appear to be supported by the data.

Overall I now support accepting this article for publication in PLoS Genetics. However, there are a few minor writing issues I would like to see resolved.

Minor concerns:

1. Lines 243-246 “In humans, only SVA, HERVK, AluS, L1, and AluY are known to be transcriptionally active…” This is a little confusing and could use a transition. In the previous paragraphs, the authors were focusing on “five TE classes/orders/superfamilies (LTR order), SVA superfamily, SINE superfamily, LINE superfamily, and DNA transposon class)…” and now appear to be focusing on specific families.

2. Line 266 is the beginning of a new results section (ie now referring to Figure 2) and the first sentence begins with “This time…” which does not really make sense in this context. The phrase “this time” typically follows some context of the “previous time.”

3. Lines 273-279: why are results now being described in terms of log2 fold change? Nowhere else are results described in this manner. This section appears to have been added in revision. It sort of comes out of nowhere, and could use an introduction/explanation.

4. Line 280 starts with “we finally” but this is not the “final” result of this section. The next paragraph begins with “furthermore.”

5. In the authors reply to comment 3 in my first review (ie reviewer 2, comment 3), they mention “The Y chromosome is richer in repeat elements than the rest of the genome but is not richer in TE insertions.” This is an important point, and the authors did add it to the manuscript (line 488-489 of discussion), but it would be helpful to state this earlier in the manuscript, ie in the introduction or results.

6. In the authors reply to comment 5 in my first review (ie reviewer 2, comment 5), they mention “Like the Y chromosome, the X chromosome could induce a redistribution of heterochromatin marks promoting TE expression (sink effect) but to a lesser extent, as observed in Drosophila.” The discussion states (line 497-498) “In Drosophila, the sink effect induced by the Y chromosome was more important than those induced by the X chromosome.” This is a fairly vague description of the X chromosome sink effect. Given this concept is an important point for the conclusions of this paper, the authors should add more detail (and references, if possible) about what is know about the X chromosome acting to redistribute heterochromatin marks (sink effect), and thus possibly promoting TE expression.

7. Also, on line 496 of discussion, it would be helpful to add a little more description of the sink effect. How is heterochromatin redistributed to the Y chromosome and how does this affect other chromosomes?

Reviewer #3: The authors have sufficiently addressed my concerns.

**Have all data underlying the figures and results presented in the manuscript been provided?**

Reviewer #1: Yes

Reviewer #2: Yes

Reviewer #3: Yes

PLOS authors have the option to publish the peer review history of their article (what does this mean? ). If published, this will include your full peer review and any attached files.

**Do you want your identity to be public for this peer review?** For information about this choice, including consent withdrawal, please see our Privacy Policy .

Reviewer #1: No

Reviewer #2: No

Reviewer #3: No

**Figure resubmission:**
---

## [Editor Report · Decision Letter 2]

Dear Dr Teoli,

We are pleased to inform you that your manuscript entitled "Transposable element expression is associated with sex chromosome number in humans" has been editorially accepted for publication in PLOS Genetics. Congratulations!

Yours sincerely,

Edward Chuong

Academic Editor

PLOS Genetics

Giorgio Sirugo

Section Editor

PLOS Genetics

Aimée Dudley

Editor-in-Chief

PLOS Genetics

Anne Goriely

Editor-in-Chief

PLOS Genetics

Comments from the reviewers (if applicable):

**Data Deposition**

http://datadryad.org/submit?journalID=pgenetics&manu=PGENETICS-D-24-00837R2

**Press Queries**

---

## [Editor Report · Acceptance letter]

PGENETICS-D-24-00837R2

Transposable element expression is associated with sex chromosome number in humans

Dear Dr Teoli,

We are pleased to inform you that your manuscript entitled "Transposable element expression is associated with sex chromosome number in humans" has been formally accepted for publication in PLOS Genetics! Your manuscript is now with our production department and you will be notified of the publication date in due course.

With kind regards,

Anita Estes

PLOS Genetics

On behalf of:
